# OpenReview forum: "From Neural Mechanisms of Short-Term Memory to the Computational Architecture of the Human Brain"
_ICLR.cc/2026/Conference — ICLR 2026 Conference Withdrawn Submission_

### Official Review · Reviewer_pCUx · 2025-10-26

**Soundness:** 2
**Presentation:** 2
**Contribution:** 1
**Rating:** 0
**Confidence:** 4

**Summary:**

In this paper, the authors focus on the problem of the case of afterimages in humans. The authors first review existing findings in detail, since the start of the first known notes on the subject by Newton. Then, they introduce a short-term memory (STM) as a potential model explaining these findings.

**Strengths:**

+ Visual neuroscience and psychology are important to the AI community in understanding how different perceptual problems are addressed by the brain.
+ The paper provides a comprehensive review on the afterimage effect.

**Weaknesses:**

Weaknesses:

Having published similar papers myself, I do not find the paper to be suitable for ICLR. As it is, it is essentially a review paper summarizing existing findings since Newton. It allocates half a page to the modeling part without sufficiently explaining or justifying the model.

Considering the recent advances in deep learning, it is not clear how/whether any of these findings/proposed architecture is relevant to the ICLR community.

I believe that a more cognitive-science focused venue might be a better fit for the paper in its current state. With a shift of focus and more content on what the findings should tell AI researchers, it might be reconsidered for a venue like ICLR.


Minor comments:

- "and AAdams et al. (2007)." => "and Adams et al. (2007).".

**Questions:**

Please see Weaknesses.

---

### Official Review · Reviewer_PLRR · 2025-10-29

**Soundness:** 1
**Presentation:** 2
**Contribution:** 1
**Rating:** 0
**Confidence:** 4

**Summary:**

This work challenges the textbook view that visual afterimages originate in the retina. The authors revive the long-forgotten phenomenon of blind spot afterimages to argue for a cortical origin. They first correlate this phenomenon with the neuroanatomical representation of the blind spot in V1-L4, identifying this layer as the neural site of afterimages. This neural persistence is then conceptualized as a form of visual short-term memory (STM). Building on this, the paper proposes a universal cortical architecture: with Layer 4 serving as STM, interfacing feedforward (L2/3) and feedback (L5/6) pathways.

**Strengths:**

- **Valuable historical review:** The paper provides a thorough and informative historical review of the "La Hire-Purkinje phenomenon" and the "Franklin effect", which is beneficial for readers unfamiliar with this specific history in vision science.

**Weaknesses:**

- **Critical Misalignment with ICLR's Scope:** This is a purely visual neuroscience *opinion article*, not a machine learning or application to neuroscience contribution. It lacks any new learning algorithm, computational model, or representation learning method, placing it firmly outside the scope of ICLR.
- **Unjustified Generalization:** The paper's universal theory for the *entire cortex* is based on a single phenomenon in *V1-L4*. This is a major logical leap, as V1-L4 is a highly specialized sensory input layer and is not representative of all cortical layers.
- **Misattributed Novelty:** The proposed feedforward (L2/3) and feedback (L5/6) architecture is not a new deduction. This structure is a foundational concept in neuroscience (the "canonical microcircuit"), which the paper fails to properly acknowledge.

**Questions:**

This work falls significantly outside the scope of ICLR. It is a purely visual neuroscience paper with no tangible contribution to the fields of machine learning or representation learning. This work may be better suited for a specialized journal in vision science, theoretical neuroscience, or cognitive science, following a substantial revision of its central claims of universality.

---

### Official Review · Reviewer_vjA1 · 2025-10-29

**Soundness:** 2
**Presentation:** 4
**Contribution:** 1
**Rating:** 2
**Confidence:** 5

**Summary:**

A visual perceptual phenomenon that we have all experienced are afterimages: the subjective impression left by something we
have just seen, e.g. a bright light or a strong reflection. The question is whether this phenomenon can inspire or at least influence the design of AI systems. The authors focus in particular on the issue of what might be the retinal or cortical origin of such afterimages. The classical view is retinal; the authors argue it is cortical. That is to say, the classical view attributes afterimages to bleaching, or saturation, of the photoreceptors in the retina. This would be relatively uninteresting for neuroAI. The authors contrast this classical view by supporting a cortical circuit origin for afterimages, based on (semi)technical arguments about positive vs. negative contrast afterimages.

The paper is a beautifully written review of the debate about origins and, I must state, included several facts of historical significance with which I was unfamiliar. For example, I was familiar with Newton's
work on color but not his work on afterimages. And I was surprised that Helmholtz did not figure more prominently in the review. Finally, relating it to Purkinjee images, etc., was interesting.

The main concern is the paper's methodology: it is basically just historical review.
Unfortunately the only vague connection to modeling and representation occurs at the very end, and is not nearly concrete enough to readily influence the neuroAI community or to inform representations for deep networks.

I think this paper would be a welcome addition to the literature on the history of visual perception,
and maybe philosophical topics connected to it. There are several journals dedicated to this, and many of the citations are to papers in those journals. While of possible philosophical relevance to the study of representations, however, it is only weakly connected.

**Strengths:**

The heart of the paper is about the retinal blindspot and the origins of perceptual afterimages. The strength of the paper is the depth of
its historical synthesis. The debate about the locus of afterimages -- retinal or cortical -- is classical, with several hundred years of
participation. The question of positive vs negative afterimages, and their relationship, was fascinating; this is mainly where the original content lies.

I agree with the author that selective visual phenomena should have an influence of whether neuroAI is succeeding, but the question is actually which phenomena and how they relate. For comparison, the current MICrONS project has produced a  "digital twin" for the mouse's visual system, with the hope that experiments can be performed "in silico" rather than "in vivo." For a general discussion, see https://med.stanford.edu/news/all-news/2025/04/digital-twin.html. The current paper is, in a sense, a call for a weaker version of this. Technically what is required is some sort of a map from the model to the biology. It is here that the author's contribution could be most relevant, but see the discussion of Weaknesses below. I recall the old question: "should robots have visual illusions?" The current paper makes the case that the answer is yes, and that afterimages are an exemplar.

**Weaknesses:**

The main weakness of the paper is the (lack of a proposed) model: it basically consists of a rough diagram (their Fig. 5). For the paper to be relevant to ICLR, it should have a serious mechanistic component, in the sense that it could (i) guide analysis; (ii) guide other researchers to consider implementing it; (iii) stimulate other researchers to check for afterimage phenomena in existing computational models of visual systems; or (iv) stimulate other researchers to build new models that might exhibit afterimage phenomena. In its current form, the paper is  too philosophical and not sufficiently computational.

In particular, the (model) diagram only vaguely captures of connectomics of V1, leaving out the many feedback pathways (e.g. long-range horizontal connections in layer 2/3) thought to be responsible for the various "filling in" phenomena. See e.g. Lateral connectivity and contextual interactions in macaque primary visual cortex, DD Stettler, A Das, J Bennett, CD Gilbert - Neuron, 2002 and references therein.  Furthermore, there is no connection to the timing of feedback, e.g. from highter visual areas or the V1 --> dLGN --> V1 cycle, or any of the other area-based cycles. I suspect that the authors are aware of much of this physiology and anatomy and it would be great to read a more detailed version of their thoughts on how, and which aspects of, this circuitry could be relevant for their conjectures.

**Questions:**

1. Is it possible to develop the diagram in Fig. 5 into more of a computational model? What would be the cell types in the different layers, and their connections, necessary to support positive and negative afterimages?

2. If a circuit were available, might it support constructive computational hypotheses about how to 'reverse' the effects of an afterimage?

3. Would the authors predict (from the putative model in 2) whether an afterimage in the shape of, say, the Muller-Lyer illusion, would still exhibit the illusion?

4. Is there a time dimension of interest? For example, saturated photoreceptors might take longer to return to normal than cortical circuitry would take to "recompute" an afterimage.

5. How might this diagram be relevant to the design of neuroAI systems; cf. the comments about digital twins above.

---

### Official Review · Reviewer_s3Pf · 2025-10-29

**Soundness:** 1
**Presentation:** 1
**Contribution:** 1
**Rating:** 0
**Confidence:** 5

**Summary:**

This paper presents no original empirical or computational work. It merely reviews existing literature and uses anecdotal observations to argue that afterimages originate in the brain, primarily referencing the blind spot as evidence. It then reiterates a standard cortical “architecture”: L2/3 feedforward, L4 short-term memory, L5/6 feedback.  This is already well-established in neuroscience textbooks and not novel.

There is no new data, model, or analysis. As such, this work is entirely unsuited for ICLR, which requires substantive empirical, theoretical, or computational contributions. This is a hard no.

**Strengths:**

This is a review paper and has no original data for me to review. As such this article is not a good match for ICLR.

**Weaknesses:**

This is not a paper relevant for ICLR.
There is no computational model, no predictions, no benchmarks. The evidence is historical, even anecdotal at places, and entirely subjective and speculative.

**Questions:**

Where is the science/quantitative data/computational validation?

**Details Of Ethics Concerns:**

This paper has nothing original or of merit. It forwards no computational model, no quantitative tests and warrants being desk rejected.

---

### Note · Authors · 2025-11-12

I have read and agree with the venue's withdrawal policy on behalf of myself and my co-authors.